# The WHO atlas for female-genital schistosomiasis: Co-design of a practicable diagnostic guide, digital support and training

Santiago Gil Martinez[1]*, Pamela S. Mbabazi[2], Motshedisi H. Sebitloane[3], Bellington Vwalika[4], Sibone Mocumbi[5], Hashini N. Galaphaththi-Arachchige[6], Sigve D. Holmen[6], Bodo Randrianasolo[7], Borghild Roald[8], Femi Olowookorun[9], Francis Hyera[10], Sheila Mabote[11], Takalani G. Nemungadi[12], Thembinkosi V. Ngcobo[12], Tsakani Furumele[12], Patricia D. Ndhlovu[13], Martin W. Gerdes[14], Svein G. Gundersen[15], Zilungile L. Mkhize-Kwitshana[16,17], Myra Taylor[16], Roland E. E. Mhlanga[18], Eyrun F. Kjetland[6,16]

1 Department of Health and Nursing Science, University of Agder, Kristiansand, Norway, 2 National Planning Authority of the Government of the Republic of Uganda, Kampala, Uganda, 3 Discipline of Obstetrics and Gynaecology, School of Clinical Medicine, Nelson R. Mandela School of Medicine, University of KwaZulu-Natal, Durban, South Africa, 4 Department of Obstetrics and Gynaecology, School of Medicine, University of Zambia, Lusaka, Zambia, 5 Department of Obstetrics and Gynaecology, Faculty of Medicine, Universidade Eduardo Mondlane, Maputo, Mozambique, 6 Department of Infectious Diseases Ullevaal, Norwegian Centre for Imported and Tropical Diseases, Oslo University Hospital, Oslo, Norway, 7 Association K'OLO VANONA, Antananarivo, Madagascar, 8 Center for Paediatric and Pregnancy Related Pathology, Department of Pathology, Oslo University Hospital, and Faculty of Medicine, University of Oslo, Oslo, Norway, 9 Ugu District Office, Port Shepstone, South Africa, 10 Department of Public Health Medicine, Faculty of Health Sciences, Walter Sisulu University (WSU), Mthatha, South Africa, 11 Instituto Nacional de Saúde–INS (National Health Institute), Marracuene, Mozambique, 12 National Department of Health, Pretoria, Communicable Diseases Control Directorate, Pretoria, South Africa, 13 BRIGHT Academy, Centre for Bilharzia and Tropical Health Research, Ugu District, KwaZulu-Natal, South Africa, 14 Department of Information and Communication Technologies, University of Agder, Kristiansand, Norway, 15 Institute for Global Development and Planning, University of Agder, Kristiansand, Norway, 16 School of Laboratory Medicine & Medical Sciences, College of Health Sciences, University of KwaZulu-Natal, Durban, South Africa, 17 Research Capacity Division, South African Medical Research Council, Tygerberg, South Africa, 18 Discipline of Public Health Medicine, Nelson R Mandela School of Medicine, College of Health Sciences, University of KwaZulu-Natal, Durban, South Africa

* santiago.martinez@uia.no

## Abstract

Up to 56 million young and adult women of African origin suffer from Female Genital Schistosomiasis (FGS). The transmission of schistosomiasis happens through contact with schistosomiasis infested fresh water in rivers and lakes. The transmission vector is the snail that releases immature worms capable of penetrating the human skin. The worm then matures and mates in the blood vessels and deposits its eggs in tissues, causing urogenital disease. There is currently no gold standard for FGS diagnosis. Reliable diagnostics are challenging due to the lack of appropriate instruments and clinical skills. The World Health Organisation (WHO) recommends "screen-and-treat" cervical cancer management, by means of visual inspection of characteristic lesions on the cervix and point-of-care treatment as per the findings. FGS may be mistaken for cervical cancer or sexually transmitted diseases. Misdiagnosis may lead to the wrong treatment, increased risk of exposure to other infectious diseases (human immunodeficiency virus and human papilloma virus), infertility and stigmatisation.

**Data Availability Statement:** Data available at WHO: https://www.who.int/publications/i/item/9789241509299.

**Funding:** The research leading to these results received funding from the European Research Council under the European Union's Seventh Framework Programme (FP7/2007-2013/ERC Grant agreement no. PIRSES-GA-2010-269245 - HNGA, SDH, BRO, FO, SGG), the University of Copenhagen with the support from the Bill and Melinda Gates Foundation (Grant agreement no. OPPGH5344 - BR, MT, SDH, HNGA, EFK), and South-Eastern Norway Regional Health Authority (SERH, Grant agreement no. 2011073 - HNGA, SDH, BRO, TGN, TVN, TF, PDN, ZLMK, EFK). The WHO Department of Control of Neglected Tropical Disease provided financial support for convening the workshop for development of the female genital schistosomiasis Pocket Atlas (PSM, MHS, BV, SMO, BR, FH, SMA, EFK). The funders had no role in study design, data collection and analysis, decision to publish, or preparation of the manuscript.

**Competing interests:** The authors have declared that no competing interests exist.

The necessary clinical knowledge is only available to a few experts in the world. For an appropriate diagnosis, this knowledge needs to be transferred to health professionals who have minimal or non-existing laboratory support. Co-design workshops were held with stakeholders (WHO representative, national health authority, FGS experts and researchers, gynaecologists, nurses, medical doctors, public health experts, technical experts, and members of the public) to make prototypes for the WHO Pocket Atlas for FGS, a mobile diagnostic support tool and an e-learning tool for health professionals. The dissemination targeted health facilities, including remote areas across the 51 anglophone, francophone and lusophone African countries. Outcomes were endorsed by the WHO and comprise a practical diagnostic guide for FGS in low-resource environments.

## 1. Introduction

Globally, at least 261 million people require treatment for schistosomiasis and up to 659 million people are at risk [1]. Female genital schistosomiasis (FGS) is a significant health problem that affects up to 56 million females in Africa. The disease is a neglected gynaecological manifestation of the parasite *Schistosoma haematobium* (Bilharzia). Schistosomiasis is categorised as one of the twenty neglected tropical diseases (NTDs) by the World Health Organisation (WHO) [2]. However, tackling FGS is a multi-dimensional problem that deals with marginalized women and girls, who face multiple and intersecting health, sociocultural, environmental, and economic challenges [3, 4]. The parasitic disease is acquired through contact with contaminated freshwater bodies used for daily chores and livelihoods, particularly among women and girls from rural or semi-urban communities in Africa. There are no point-of-care tests that could systematically support the WHO policy to screen-and-treat FGS [5, 6]. The clinical examination is made using a speculum [7] and should be supported by concurrent colposcopy. Colposcopy is done by trained health professionals, usually gynaecologists. The colposcope is expensive and not readily available. The lack of appropriate training, clinical instrumentation and knowledge, results in a large proportion of FGS patients undiagnosed, untreated and without access to preventive measures [6]. FGS may be misdiagnosed as a sexually transmitted infection (STI) or cancer. The wrong diagnosis may result in incorrect treatment given to girls and women, and may result in marital discord, blame, stigma, and violence [8–12]. FGS is a risk factor for human immunodeficiency virus (HIV) and, possibly, also for human papilloma virus acquisition [7, 13]. There is an urgent need for intervention supported by the release of a United Nations AIDS's (UNAIDS) and World Health Organisation's (WHO) public document [14] to raise awareness about schistosomiasis, FGS and their potential effects on Reproductive health and HIV vulnerability. These effects are particularly dangerous among adolescent girls and young women in sub-Saharan Africa, who are also among those most affected by the HIV epidemic [15]. Lastly, tropical diseases have started to appear in different latitudes. An FGS outbreak identified in Corsica (France) in 2014, showed that the risks of snail-borne diseases in Europe might have been overlooked [16, 17]. These risks are contemporary to the urgent calls on changing the way researchers think about disease transmission in the times of the global warming [17, 18].

### 1.1 Lack of FGS diagnosis

FGS diagnosis requires comprehensible knowledge by the health professionals. There has been reports on FGS and its epidemiology for over 130 years, which provides the knowledge on

gender-based and exposure-related clinical manifestations [19]. Systematic sex-and-age-appropriate patient information, and prevention and diagnostic approaches are therefore needed [20, 21] However, the development of a professional diagnostic competence requires several weeks of bedside training and exposure in endemic areas [22, 23]. A systematic training has not yet been implemented. Furthermore, issues that are typical for the FGS patient population (e.g., transport, gender-specific parental control and husband control, cultural and potential seasonal effects) should be considered in the design of new diagnostics and education for both primary and specialised health care [24–28]. Health professionals who currently provide vaginal speculum examinations could be trained to reduce the risk of misdiagnosis [26]. Training should encompass recognition of the manifold lesions, including subtle cases, treatment, index case information, tracking of related cases and community management [6]. In this context, education and training programs designed with patient safety, clinical outcomes, evidence-based point-of-care diagnosis, adequate patient information and education and practical implementation in low-resource settings may enhance the effective diagnosis of FGS.

To tackle the challenges described above, an international group of scientists and health specialists aimed to co-design a practical standard for the diagnosis of FGS in low-resource settings typical of rural areas. The method employed in these workshops was "co-design" [21]. Co-design is defined as "the creativity of designers and people not trained in design, working together in the design development process" [29]. Co-design belongs to the human-centred design research, as practiced in the design and development of products and services. The principles of a human-centred design of an interactive system refer to the explicit understanding of users, tasks, and environments, involving the users in an iterative process of design and evaluation.

## 1.2 Human-centred design (co-design)

The co-design method was developed by the European and Nordic School of Participatory Design [30]. Participatory design provided the means "through which ordinary people at their workplace can more democratically promote their own interests" [31]. Co-design is used across disciplines and contexts. Particularly in health, co-design has been emergent in the 2010 decade, with a noticeable example such as experience-based co-design (EBCD) [32]. EBCD is a method involving users of healthcare services and understanding their experiences of those services, identifying improvements, and working together to make the changes happen. It is generally divided into four steps: capture, understand, improve, and measure. Together they represent an example of a method based on patient-centred care [33, 34]. Preparation, allocation of resources and training are key factors to enable the appropriate use of a co-design method in healthcare [33]. The process that co-design provides can be transformational for participants (health professionals and patients), although it is hard to quantify [35]. Co-design has also been used in low-resource settings in public health community-based participatory research [36]. More recently, a co-design control intervention for schistosomiasis was implemented for school-aged children in Zanzibar [37]. It resulted in education, training, development of activities and events, and the installation of urinals and laundry-washing platforms. The study was part of the Zanzibar Elimination of Schistosomiasis Transmission (ZEST) alliance formed in 2011 [38, 39] and it suggested that the approach would facilitate sustainable and effective interventions.

The problem addressed in this research deals with how to transfer appropriate knowledge to healthcare professionals in FGS endemic areas, especially rural areas that have minimal or non-existing laboratory support. Based on reported co-design experience within health (see EBCD, [32] and rural areas with schistosomiasis [37], co-design was chosen as a method for

transferring the knowledge to healthcare professionals, who were stakeholders not necessarily trained in design but could work together towards a shared goal. In a series of workshops, experts and users assembled existing knowledge, such as clinical procedures for health professionals, parasitology, and pathogenesis to correctly diagnose FGS, avoid misdiagnosis, improve the index of suspicion, and educate the next generations of health professionals. The next sections describe the material and methods used in the co-design workshops and their outcomes. In these workshops we sought to provide a medical booklet and bedside poster with examples of FGS lesions, explore the requirements for a diagnostic support for FGS and e-learning for health professionals. Finally, we discuss the implications and limitations of this research.

## 2. Methods

### 2.1. Ethical considerations

The research was conducted under the principles expressed in the Declaration of Helsinki [40] for human research, acknowledging that rural women are vulnerable and subject to perceived ownership by husbands/in-law families and stigmatised for the consequences of FGS [21, 41]. The images were obtained in four countries (Malawi, Zimbabwe, South Africa and Madagascar). The images were subsequently anonymised and not identifiable. Images and specimens were sampled from sexually active women between 15 and 49 years of age. Colposcopic images of other diseases were included for differential diagnostic purposes. See [42] for more detailed information. Participation was voluntary and, in the consent procedures, the participants were informed that they could withdraw their consent at any time without penalty and that they could request their material (e.g., images) to be consequently destroyed.

The anatomy and details depicted in the images do not provide basis for identification of the origin, person of the capturer. Permission was granted by the authorities and ethics committees in each country. In Zimbabwe, the Provincial and District Medical Directors, the village chief and village meetings gave their permission to conduct the study. Ethical approval was given by the Medical Research Council of Zimbabwe and by the ethical committee of the Special Program for Research and Training in Tropical Diseases Research, United Nations Development Program/World Bank/World Health Organisation (UNDP/WB/WHO). In Malawi, ethical approval was given by the Medical Ethical Committee of Malawi, Ministry of Health and Environmental Affairs 1993 and by UNDP/WB/WHO Special Program for Research and Training in Tropical Diseases (WHO TDR). In South Africa, three ethics' committees granted permission to perform the study: Biomedical Research Ethics Committee Administration, UKZN in KwaZulu-Natal, Department of Health, Pietermaritzburg, KwaZulu-Natal (KZN), Regional Ethics Committee (REC) Eastern Norway, and the European Group on Ethics in Science and New Technologies 2011. The Departments of Health and Education in KZN gave local permission. In Madagascar, ethical permission was obtained from the Committee of Ethics at the Ministry of Health in Madagascar.

In all countries target populations were sexually active women of reproductive age only. Study information was provided to the target populations in the local languages for the image collection. Participation was voluntary and participants could withdraw at any time without penalty. Cultural appropriateness was ensured and tested by women of the same age group and culture working in the respective research institution to specifically address any language or cultural sensitivities. Consent procedures were registered in written form (or thumb print when necessary), except in Zimbabwe, where the consent was orally registered. The consent process followed these steps: the WHO contacted the ministry of health of each country; then the ministry of health appointed specific people to join the process of development of the WHO pocket atlas. The consent was documented and witnessed by the appointed people. The

institutional review board (IRB) approved the overall study procedures and consent process. In all cases, the aim was to maximize autonomy and dignity for study participants. Following consent, all willing, sexually active women, aged 18–49 years, were offered gynaecological examination by health professionals with expertise in FGS. Consent was re-confirmed by the physician before each step of the investigation. If a patient did not give their consent, no image would be taken. Treatment and follow-up for schistosomiasis, sexually transmitted infections (STIs), cancer and other conditions were given in all sites.

Workshop participants were not named, only collaborative contributions were presented. For the co-design workshops, all participants signed a written consent and were orally informed about the aim of each workshop, the project overall aim, facilitators' name, position, and responsibilities. The participation was voluntary. Participants could leave the workshop at any time without reason. Neither apologies nor withdrawals were registered. The individual featured in Fig 3 has provided written informed consent, as outlined in the PLOS consent form, to have their image published alongside the manuscript.

## 2.2 Data collection

There was one data image collection followed by a series of HCD workshops. The data image collection consisted of 10 000 images taken from 3272 participants from three different countries during a colposcopy examination: Malawi (52 participants), Zimbabwe (700 participants), Madagascar (120 participants), South Africa (2400 participants). These countries were chosen as data collection sites because they were intentionally different in terms of age and endemicity. All the data collected from these participants followed the ethical procedures described in the previous Ethical considerations section. The camera was an integral part of the colposcope [43]. In consenting women, images could be taken seamlessly during the examination. The focus range was narrow, as it has been described by [44] and some women had multiple lesions, some at different "depths", e.g transformation zone of the cervix, in the fornices and on the vaginal wall [6]. Therefore, if the lesions were at different depths, multiple images were taken. There were several challenges faced in obtaining quality images. For instance, lighting, where the tilt of the vagina made it sometimes difficult to do inspection of anterior and posterior surfaces; when a patient moved during the procedure, the images would get blurred and repeated, and many of these issues were not observed until the images were processed. This issue made challenging to identify every single image identification number with its corresponding patient identification number. The images are not yet available, but the authors have applied for funding to make a database in liaison with the WHO.

The workshops took place in the University of KwaZulu-Natal—Nelson R. Mandela School of Medicine, South Africa. They followed a Human-Centred Design (HCD) methodology [45, 46], as shown in Table 1. The problem descriptions (HCD 1) are presented in the introduction and the agreed outcomes (HCD 2) are presented in the Methods chapter. In the Results chapter, we present co-design, plenary presentation and debrief (HCD 3–5). Three workshops were held on FGS, and three topics were addressed, namely (1) the making of a pocket atlas for clinicians, (2) point-of-care diagnostic support and (3) training of health professionals. The latter two topics were addressed in both last workshops. However, for practical reasons and clarity, the three topics are presented separately.

Table 2 shows that the workshop participants were recruited from health and academic institutions, local authorities, health professionals, patients, relatives, local population, and formal and informal networks connected to the research consortium. For instance, in South Africa the participants were recruited in collaboration with the Department of Health and Department of Education, to approach the community and identify relevant stakeholders and/

**Table 1. Description of human-centred design (HCD) phases.**

| HCD phase | Name | Description |
|---|---|---|
| 1 | Problem description | Patient and health professional journey: presentation, bottle necks, gaps, opportunities |
| 2 | Agreed outcomes | Identify and agree on desired outcomes: clinical, patient, epidemiological, social |
| 3 | Co-design | Multi-stakeholder group co-design paper-based prototypes, aiming at agreed outcomes |
| 4 | Presentation | Presentation in the plenum of each group's proposed solution, addressing questions from the audience |
| 5 | Debrief | Key points and highlights shared among participants. Outline of way forward |

or gatekeepers. Several information was organised with these stakeholders where they could ask questions. Then, they were invited to consent procedures where they could also ask questions. Participants were distributed into different pre-defined groups in all the three workshops. The main criteria for group distribution were to include at least one participant from each stakeholder group, when possible, to ensure diversity. Stationary such as flip-overs, markers, colour pencils, paper and post-it, which were used by participants in the co-design workshops to sketch and represent their ideas.

## 2.3 Definitions

**2.3.1 Disease characteristics.** FGS was clinically defined by mucosal lesions. One major lesion was having the so-called sandy patches. The grainy sandy patches were defined as having grains approximately 0.05 mm by 0.2 mm long, shaped as minuscule rice grains, appearing singly or in clusters of up to 300 grains [47]. The homogeneous yellow patches were defined as sandy looking areas with no visible grains when using a 15-times magnification setting on the colposcope. Rubbery papules were defined as papulous lesions, firm as hard rubber [48].

**Table 2. Participants in co-design workshops on FGS categorised by stakeholder.**

| Topics | WHO pocket atlas | Diagnostic support/Training of Health Professionals | |
|---|---|---|---|
| **Meeting date** | **04/10/2014** | **01/04/2019** | **09/01/2020** |
| **Duration (days)** | **6** | **1** | **1** |
| **Participants*** | | | |
| WHO representative | 1 | | |
| National health authority | 3 | | |
| FGS experts (researchers) | 6 | 3 | 5 |
| Gynaecologists | 7 | 1 | 2 |
| Nurses | 3 | 1 | 2 |
| Medical doctors | 6 | 3 | 4 |
| Public health experts | 4 | 1 | 1 |
| Professors | 6 | 4 | 5 |
| Schistosomiasis experts | 4 | 2 | 3 |
| Technical experts** | 3 | 3 | 2 |
| Members of the public | 0 | 0 | 3 |
| Total participants | 43 | 18 | 27 |

*may be categorised in several positions, e.g. several medical doctors were also professors

**ICT experts, digital technology

Abnormal blood vessels were defined as pathological convoluted (corkscrew), reticular, circular and/ or branched, uneven-calibre blood vessels visible (by 15 times magnification) on the mucosal surface. Contact bleeding was defined as fresh blood originating from the mucosal surface. Pre-contact bleeding was defined as darkened blood on the mucosal surface in the absence of recent or present menstruation.

**2.3.2 Health provider roles and clinical procedures.** A health professional in this manuscript is defined as a nurse, midwife, general practitioner, or gynaecologist. We differentiate between those who have been trained to use speculums for intra-vaginal inspections and those who do not. Most of those who use speculums have not been trained to inspect the vaginal walls. Furthermore, usually only a sub-group of gynaecologists regularly use a colposcope, the instrument used in cervix cancer investigation.

## 2.4 Materials

The medical instrumentation used for collection of images were a speculum and photo colposcopes, which included the Olympus OSC 500 Photocolposcope (Olympus America Inc., Center Valley, PA, USA) with a mounted Olympus E420, 10 megapixels (Mpx)(Olympus America Inc., Center Valley, PA, USA) or a Leisegang Photocolposcope, (Script-O-Flash;Leisegang Feinmechanik-Optik GmbH, Berlin, Germany) with a mounted Canon EOS 350D and 500D, 10 and 18 Mpx respectively (Canon Inc., Tokyo, Japan). Eyepieces, lamps, bulbs, and surrounding light conditions were appropriately adjusted to optimise data collection, often needing to use a magnification of 15 times or more. The continuously micro-meter zoom of the Olympus was used. Images of the lesions were printed as A6-size photographs.

The workshop's participant's presentations were filmed. The tangible results of the workshops were made by paper prototypes supported by props and oral presentations (audio and video), all made by the participants themselves. The outcomes of the workshops served to illustrate viable products and the fact that education and training were part of ongoing research and funding search connected to the research consortium. The paper outputs of the workshops were photographed and kept as records. Their meaning was described in the text.

## 2.5 Data analysis

Before the workshop, two FGS experts (EFK and HGA) sorted the 10 000 images by lesion type and quality. The positive cases (e.g., registered diagnosed as FGS) and look for images that show good-quality observable lesion. In total, they selected 28 images were chosen for each of the four FGS lesion types. Images were pinned on four trays, one tray for each lesion. They were then discussed, analysed, and selected by FGS experts, gynaecologists, and other health professionals. A legend was given for each image. Disagreements were discussed until a consensus was reached.

Workshops 2' and 3's videos were qualitatively analysed using content analysis [49]. MS Sharepoint automatic video transcript and caption was used to produce the initial data. The transcripts were moderated by at least two of the authors. Disagreements were discussed until a consensus was reached. A disagreement would require re-analysis of the video presentation by a third author. Analyses reached consensus in all the video presentations. The analyses comprised the coding of the transcription and their derivation into categories and subcategories (presented in the following section).

## 2.6 Inclusivity in global research

Additional information regarding the ethical, cultural, and scientific considerations specific to inclusivity in global research is included in S1 Checklist.

## 3. Results–co-design, presentation and debrief

### 3.1 Co-design: Atlas and poster for gynaecological examination

The co-design workshop goals were:

A.  Review and choose suitable images from the repository of anonymised images.

B.  Produce the text for inclusion in the pocket-size atlas.

C.  Make a prototype in English and draft translations to French and Portuguese languages.
    In addition, two more goals without allocated resources were developed ad-hoc:

D.  Determine the Atlas product development plan and suggest a dissemination strategy.

E.  Suggest advocacy and resource mobilization in support of ongoing work efforts on FGS issues.

In the making of the WHO Atlas, the co-design and presentation phases (HCD 2–4) were repeated several times, circulating images, texts, and prototype sections between the groups, letting participants review each other's work. Discussions were interrupted by plenary discussion whenever needed. All the steps were collaboratively performed to accomplish the described goals.

The image selection followed a four-stage blinded process to review more than 10 000 photocolposcopic images. The images selected were those with characteristic FGS pathology. The final set of 28 images presented in the Atlas were from Madagascar, South Africa, and Zimbabwe. Individual consent was obtained in each case. Personal identification markers were removed for each image after selection. It was therefore not possible to identify the origin of each image, and for the same reason the person who captured the individual clinical data was not identified. Diagnoses were given to study participants during investigations and images were take [7, 26]. FGS positive cases were preselected, and images reviewed on a computer screen (by FGS experts HNGA and EFK). Approximately three hundred images were printed, and a further selection process was done, by the same experts, to only include those images that were visible on 10.5 x 14.85 cm (A6) size images.

The participants were given a summary of the clinical findings, the latest research on FGS and later assigned to two working groups led by an expert in FGS. The experts had pre-selected 60 images of the types of lesions of FGS: (1) sandy patches, (2) homogenous yellow sandy patches, (3) rubbery papules and (4) abnormal blood vessels.

Each group reviewed two disease manifestations (30 images, see Fig 1) and selected the (subjectively) clearest and most representative, wrote a brief legend, and suggested symbols (e.g., arrows, circles) as visual support.

Afterwards, the groups exchanged image collections and texts, reviewed each other's work, and made changes. Plenaries were held when needed to discuss issues before going back to the images and achieve consensus (Fig 2). A draft version of the prototype (with images, arrows to point to lesions and draft legend) was made for review. After several review rounds, a total of 28 images were selected with their figure legends, reviewed five times by the groups. Furthermore, supplementary text was written on symptoms, investigation methods and treatment. This was reviewed in the same way as images and included in the prototype booklet.

The main outcome of the workshop was the production of the FGS Atlas prototype (before WHO formatting) depicting lesions characteristic of FGS (Fig 3). Further resources were created during the workshop: (1) a printed prototype or poster for dissemination to all African countries' endemic of schistosomiasis; and (2) a dedicated web page. The Atlas draft (before WHO formatting) together with the following items were produced during the workshop in

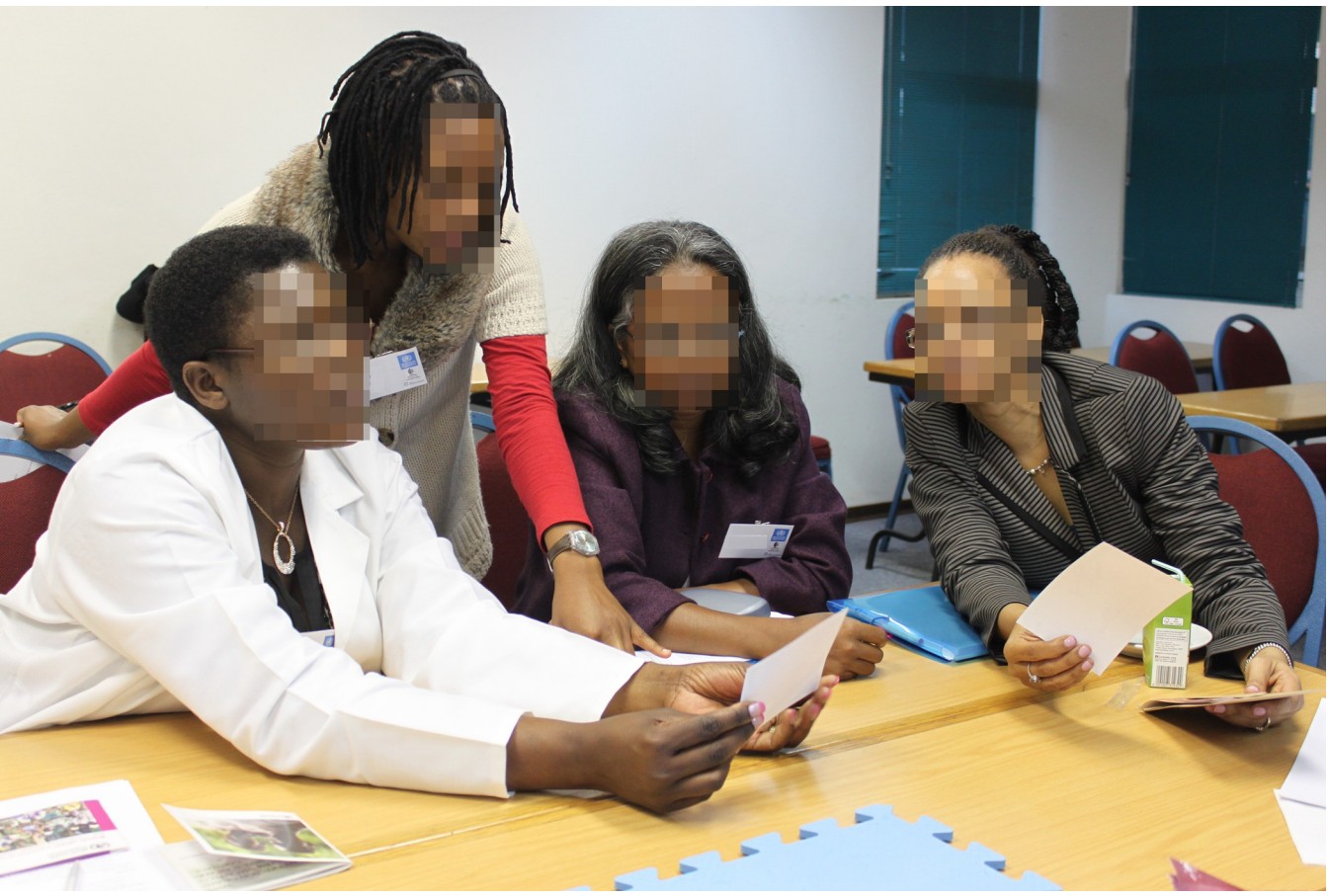

**Fig 1. FGS Atlas image selection.** A group working on the selection of images for disease manifestation.

English, French and Portuguese languages and included a space for e.g., ministry of health logos.

Below is the list with all the outputs of the workshop:

A.  Atlas for printing, 28 images, CMYK colour space (see Fig 4).

B.  Bedside poster: 2 examples of each lesion, CMYK colour space.

C.  Atlas for viewing on monitor in full resolution, RGB colour space.

D.  Atlas for viewing on web in low resolution, RGB colour space.

E.  Microsoft PowerPoint presentation with images from the Atlas, for use in promotion and training.

F.  Dedicated Web page material: information to the public, clinicians, and relevant stakeholders.

In addition, during plenary sessions, participants discussed during plenary sessions the transfer of knowledge to train health professionals in FGS diagnosis as shown in Fig 5. It is important that the referral institutions (gynaecologists, colposcopists) are ready to manage patients with FGS. They should be more knowledgeable about the patient's primary contact with the health system.

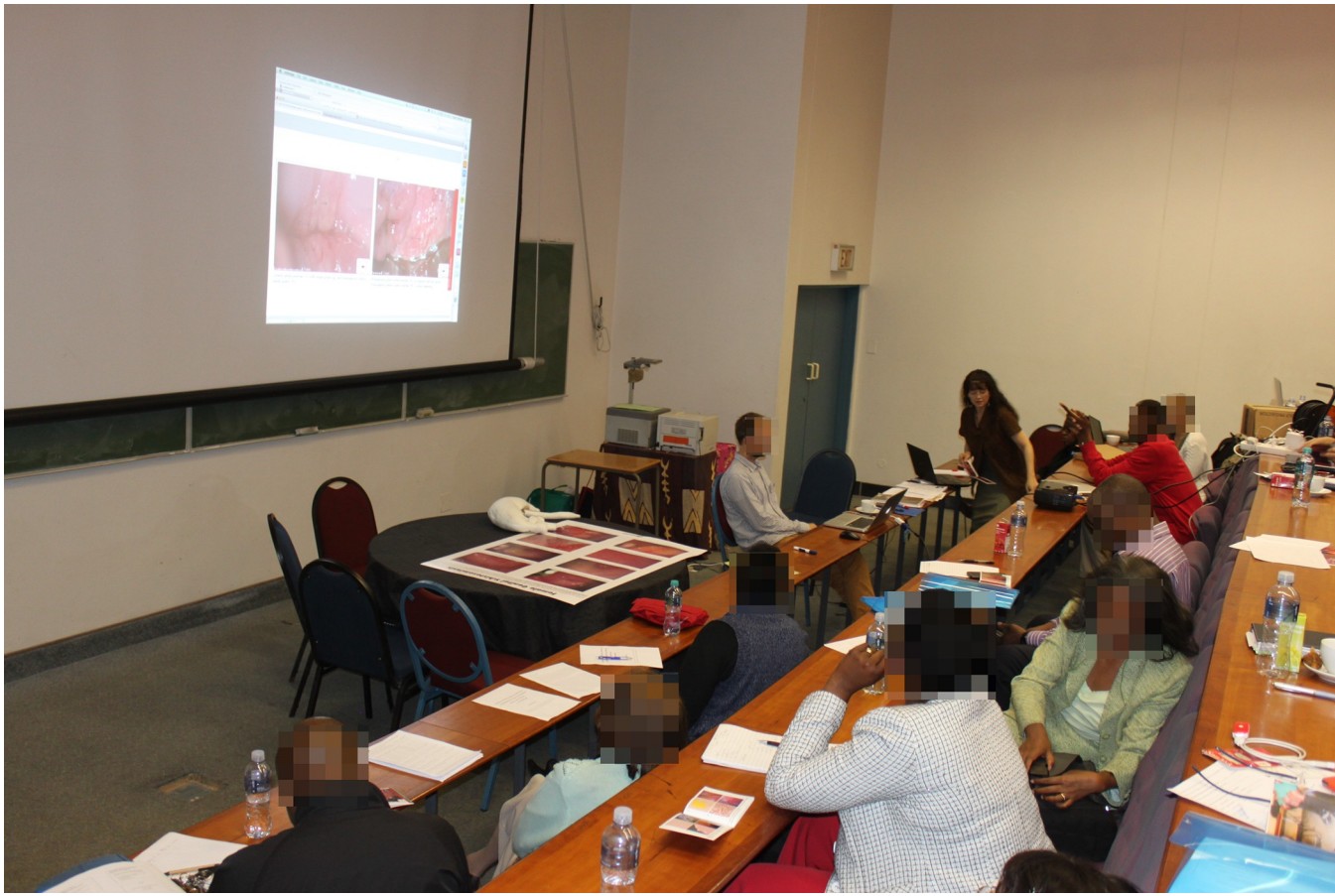

**Fig 2. Plenary discussion.** Image selection and significance discussion in the plenary with all the groups present.

### 3.2 Co-design of diagnostic digital support for health professionals

The participants (Table 2) were given a comprehensive presentation of the FGS disease, life cycle, infectious pathway, and examples of lesions. They were distributed in groups and received an explanation of the 5-phase HCD method. The groups presented their work in the plenary sessions.

The aim was to discuss the content and structure of the digital diagnostic support mobile application to be used by health professionals at the point-of-care (POC).

It was unanimously suggested by the groups to design a device suitable for low-resource settings and with low digital-literacy skills. It was followed by a discussion on the exploration of different alternative technologies. The device would be connected to a smartphone and then the relevant personnel would be able to analyse and interpret the images taken by a single screen touch (Fig 5).

The device should be utilisable for both FGS and cervical cancer, and ideally, also for other diseases. A trained health professional should be able to make a diagnosis supported by the technology. The device should be able to diagnose 85–90% of the cases, which would contribute to a reduction in the number of referrals (Fig 6).

The designer should consider that many endemic settings have unstable electrical supply (technical requirements: chargeable/uninterrupted power supply (UPS)), budget to purchase and operate the equipment, gynaecological bench for examination with speculum, and

**Fig 3. FGS Atlas prototype.**

smartphones and a smartphone application (app) for support. A seamless triage could be developed to screen for potential FGS patients as they come to a clinic to do the examination where applicable. For the instrumentation, the device would ideally have a camera coupled with a light source. The clinician should have one hand free while performing the examination. Telemedicine should be then part of the design to perform quality control and provide online

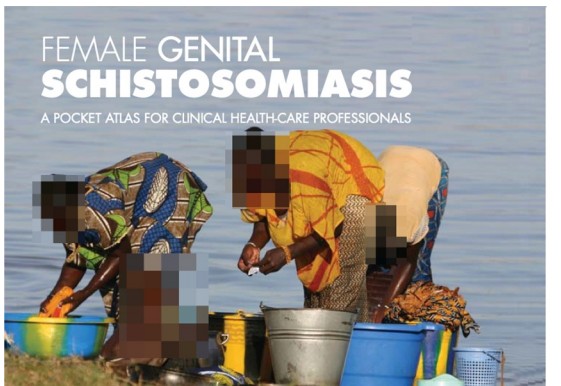

**Fig 4. The WHO FGS atlas.** Cover and third pages [50].

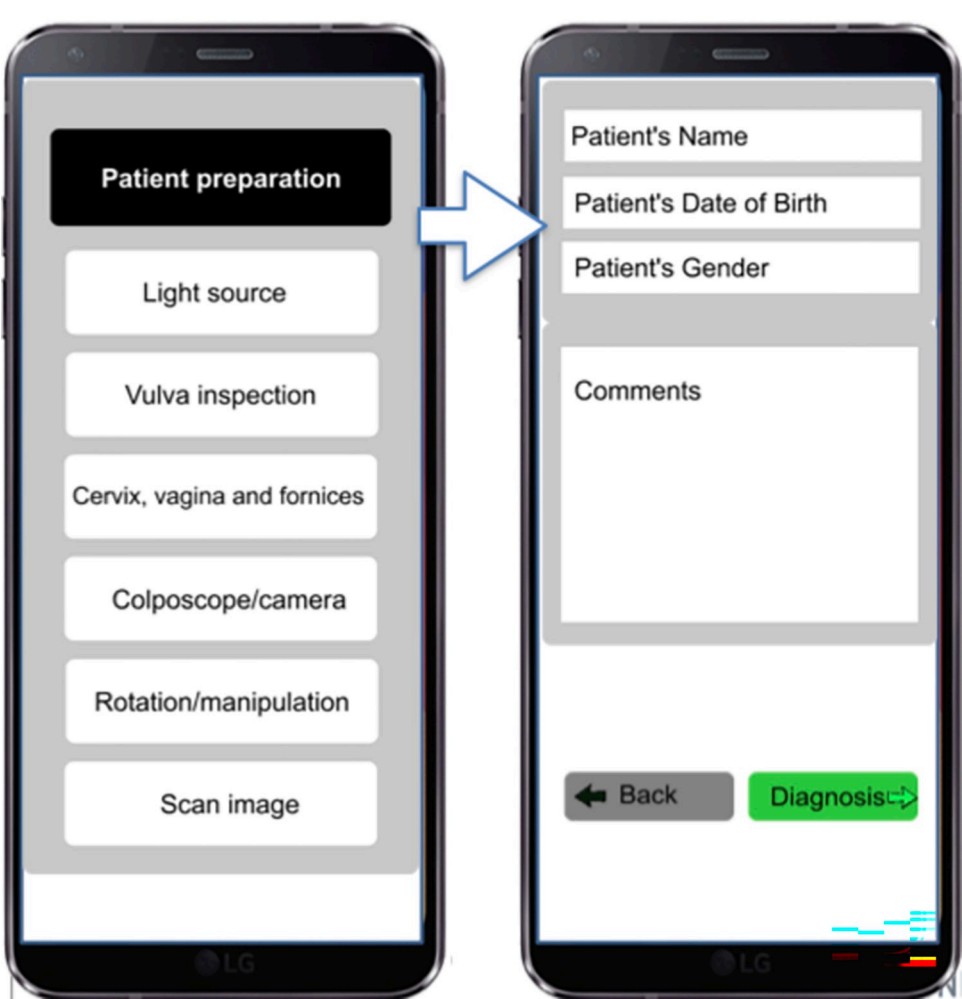

**Fig 5. FGS diagnosis support: functionalities.** Co-designed example based on groups outputs.

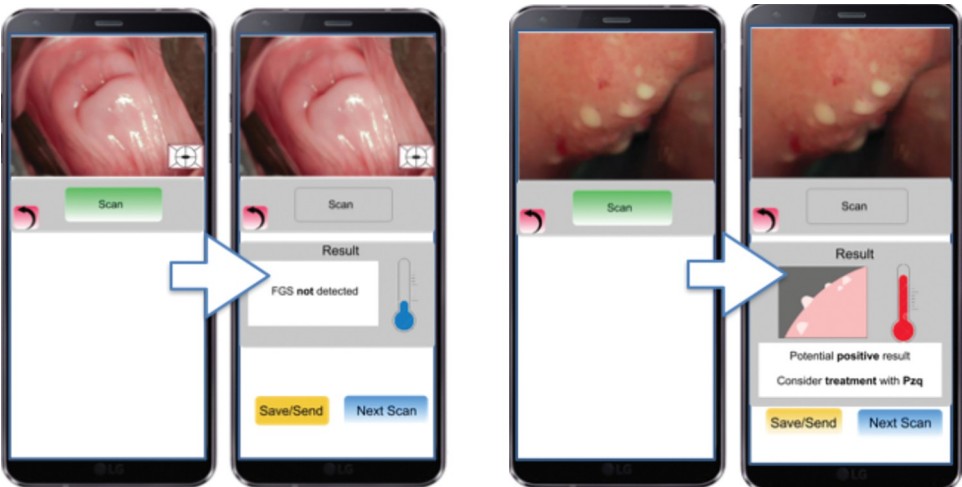

**Fig 6. FGS digital diagnosis support: FGS detection.** Co-designed example based on groups outputs.

assistance from a knowledgeable gynaecologist/trained person at the other end. A system should be in place to ensure patients are notified about their results. An example of how such a telemedicine system in low-resource settings has been used elsewhere, with reported perceived benefits [51]. Store-and-forward networks, i.e., providing a network template which can be used without technical expertise, were clinically useful, sustainable, and potentially cost-effective [52]. Efforts for the implementation of such system are ongoing in a research project that follows this manuscript, and it is funded by the European Union. In it, three study sites will send high-quality images to a team of experts to evaluate the quality but also the clinical decision making associated with each diagnosis. Ethical considerations and data protection procedures/technical solutions will be in place, complying with the General Data Protection Regulation in Europe (GDPR) and its South African (and other countries) counterpart (e.g., Protection of Personal Information Act No. 4 of 2013). Patients should be informed, and consent is important. In case images are captured, patients should be informed that a picture of the interior of their genitals will be taken and might be sent somewhere else. In these cases, it is essential for the patients to understand why the picture was needed.

A rich discussion was triggered around the topic about seeing the cervix and all the vaginal walls or just the cervix and its surroundings. Due to the difficulties of inspecting the vaginal walls, it was suggested to examine the surroundings of the cervix, as studies indicate the same density of FGS eggs in the surroundings of the cervix as in the vaginal walls. However, the audience expressed doubts about the uniform density of eggs in the cervix and surroundings and the rest of vagina, given that eggs could come from different worm couples. This means that there could be more than one pair of worms laying eggs and therefore the density could differ depending on where the eggs were laid.

It was suggested that FGS could be diagnosed using the "hysteroscope", an operative telescopic camera for the uterus, with a diameter of 5 mm, and an external monitor for visualisation. The instrument could show a few sections of the cervix with enough magnification, without the need of the speculum. The hysteroscope has a light and could provide a high magnified picture. The high-quality images would be digitally and securely sent to an expert and/or automatically analysed. However, it cannot be rolled out to the rural clinics because it is highly technological, expensive, and not generally available. Nevertheless, it should be explored as a prototype for simplification and scale-up. The audience asked whether the hysteroscope would enable the examiner to see the vaginal walls or whether that requires the use of a speculum. The response was that beyond the cervix and its surroundings, the hysteroscope cannot provide further visual information. A speculum would then be needed. The audience also raised questions about the safety of the device. A long instrument that could reach the uterus with the consequent risk of perforation could be dangerous to use in low-resource areas where health care workers lack appropriate training. They recommended a shorter device for primary health care professionals.

Current speculums do not allow the examiner to see all the vaginal walls at once. Transparent speculums might seem to be prone to reflections. Clinicians usually solve the problem by rotating the speculums, in a way that the angle allows to see the vaginal walls from all perspectives. This increases the time per consultation, which might not be well received by healthcare professionals that are under stress. In addition, to get to see all the possible angles the clinician needs to change sitting positions several times, improvising, because the gynaecological investigation bed cannot always be moved up or down. Lastly, one member of the audience (representing citizens and patients) expressed concern about somebody taking a picture of private parts and what could be seen on a screen and be posted in social media.

The presenter responded that for the public to know about the lesions in the vagina was going to be limited because it is difficult to recognise even in an operation theatre. An FGS

Table 3. Content analysis summary of the diagnostic digital support.

| Code | Category | Theme |
|---|---|---|
| FGS is not currently diagnosed because is difficult to see without colposcope (naked eye) | It takes several weeks of bed-side training for healthcare professionals who use a colposcope | Need for evidence-based diagnostic clinical support |
| FGS diagnosis workflow: expertise, experience, instrumentation | Ideal versus real clinical workflow | FGS clinical protocol for low- and high-resource settings |
| Portable affordable device, able to scan visible and invisible light spectrums for automatic image recognition and lesion identification | Technology that enables diagnosis improvement | Low-cost technology-enabled healthcare usable diagnostics |

expert answered that the Atlas does not have many vaginal wall pictures because they are the hardest to take with enough quality to be meaningful. The current equipment is not good enough and many pictures appear unclear. A new tool such as the one described in the presentations might make it possible, arguing that lately more lateral inspections are successful, such as when stents are placed in coronary arteries, for example using the optical coherence tomography (OCT) technology. The new diagnostic tool should provide a better way of capturing information and consulting for a second opinion. Other similar procedures in terms of visual information, such as gastroscopes or colonoscopies, use air to ensure that the tissue is expanded (e.g., colon), but this is difficult for vaginal walls which are initially collapsed.

Another concern expressed by a member of the audience had to do with the ethical approval for taking pictures of vaginal walls and cervix. The research team responded that the highest standards of ethical and data protection approvals were obtained in all the countries where the research was performed. The images would be taken as part of a routine examination, being safely stored in an electronic health record system for documentation purposes. The storage of images was relevant to ensure quality control and second opinion. Lastly, an FGS expert mentioned previous efforts of using digital image analysis which reached a sensitivity of 60–83%, and a specificity of 70–73% depending on the device quality. This may be insufficient for clinical practice, requiring a minimum of 80–90% of sensitivity.

A summary of the points mentioned above are shown in Table 3.

## 3.3 Co-design on awareness and training for health professionals

The study participants and settings were the same as the ones described in the previous section. The aim was to identify key areas in which health professionals require more training and awareness and how the training could be structured.

The workshop participants indicated that there was not a standardised system or structured guideline for FGS teaching. Participants agreed that the situation may contribute to misinformation about the disease and misconceptions. Special emphasis was made on the training for primary healthcare nurses, who were considered the entry point of many patients into the healthcare system. Other professions were mentioned, such as laboratory specialists, who receive specimens from endemic areas. They should analyse for schistosomiasis in genital specimens even if such analyses were not specifically requested. Training would be essentially, evidence-based, with the latest advances in knowledge being integrated in the practice via the diagnostic practicable standard and an up-to-date management protocol.

Current practice does not include management of FGS in rural clinics, family doctor rooms and gynaecologists' rooms, as observed and/or experienced by several members of the groups.

An ideal workflow for FGS diagnosis was described by the groups: a clinician will interview a patient as they come, taking history, and asking about symptoms. As per standard procedure,

it is followed by measuring blood pressure and glucose levels. If there are genital symptoms or infertility, patients should undergo gynaecological examination. If the problem cannot be solved by the clinician the patient will be referred further (to secondary, tertiary, or quaternary institutions). In most primary health care clinics in South Africa, the primary health care nurses are the first point of contact with all the patients, then a doctor will come at regular intervals to manage difficult cases and certain patients with chronic conditions. In the instance of a patient with FGS, typical symptoms that should raise the index of suspicion for FGS would be abnormal discharge, bloody discharge, infertility, and dyspareunia. The patient should be asked about exposure to freshwater bodies and, if they have not relayed it already, they should be asked about a history of vaginal discharge, pain during urination, and blood in urine ("red urine"). Then, if any of these questions was positively answered, the health professionals may test the urine by dipstick or send to a laboratory requesting microscopy for Schistosoma eggs and red blood cells in urine. The participants noted that bloody urine is not a good test in adult pre-menopausal females.

Many nurses in schistosomiasis endemic countries perform gynaecological examination using a speculum. However, nurses do not do colposcopy, this is done by gynaecologists. However, all health professionals lack training in FGS, the workshop participants indicated that health professionals are not aware of the morbidity caused by FGS in the cervix, vagina, and vulva. The workshop participants expressed that the FGS Atlas should be available in local languages.

A summary of the points mentioned above are shown in Table 4.

**3.3.1 Post-graduate training.** In KwaZulu-Natal, training institutions may only hear about FGS in the form of a list of differential diagnoses. Furthermore, it is not mentioned in the treatment protocols for hospitals or primary health care. The workshop participants suggested that colposcopy courses should include FGS, where course organisers should ensure that they get should get extra Continued Professional Development (CPD) points for passing a test in FGS diagnosis. Health professionals who already use a speculum could do an e-learning course in FGS diagnosis, as part of their CPD. The content should be based on the WHO pocket Atlas for FGS. For those who have a colposcope (handheld or mobile), additional information should be made available on technique to explore the entire vaginal surface. Training would include ethical and data protection aspects such as patient information and informed consent.

**3.3.2 Graduate training.** The current curriculum for medical students and nursing students does not yet include FGS. FGS should be mentioned as a differential diagnosis for STIs and other reproductive tract diseases. It was agreed that the curriculum should include the transmission cycle. Training should be integrated into didactic lectures, educational material (poster, videos), online (Fig 7) and offline (FGS Atlas). It should provide cues to increase the index of suspicion, examples of history taking, clinical information for the patient and for

**Table 4. Content analysis summary of the e-learning for health professionals.**

| Code | Category | Theme |
|---|---|---|
| FGS is a neglected tropical disease endemic in most African countries, especially in low-resource rural settings. | FGS is a differential diagnosis for sexually transmitted diseases and cervical cancer, often mis-diagnosed and wrongly treated. | Gaps in knowledge of scholars and training of health professionals are determinant |
| From primary health care to tertiary health care FGS is currently not diagnosed, often reported as an incidental finding. | FGS is absent in the curriculums of medical and nursing schools. FGS patients are scattered and represent index case patients. | |

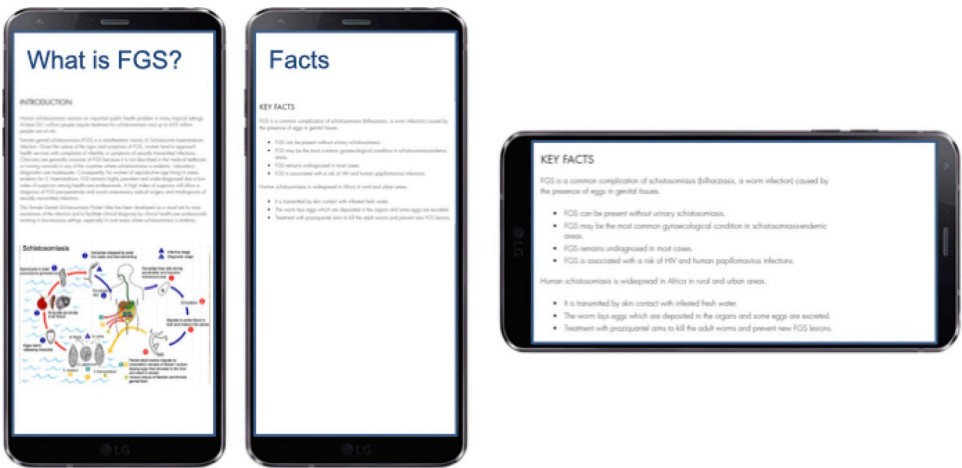

**Fig 7. FGS e-learning.** Co-designed example based on the groups' outputs shown on a smartphone interface.

referral institutions, visuals of FGS lesions and differential diagnosis. It should be explored if mannequins can be used to facilitate the recognition of lesions and the investigation procedure of cervix and the entire vaginal surface [1]. The use of high-fidelity mannequins would allow for trial-and-error experiential learning with the different kind of lesions before going into the field. It is therefore in the agenda of the authors for future research, considering competence-based evaluation to assess its effectiveness and clinical skills.

All the training and education initiatives discussed could potentially face implementation barriers, due to costs (e.g., instrumentation, learning material), accessibility (transport to nearest higher education institution or healthcare facility, Internet connectivity, data packages) or cultural factors. However, these challenges are belittled by the fact that the direction of training requires several weeks.

**3.3.3 Professional groups that do not do intra-vaginal inspection.** Laboratories receiving specimens from endemic areas should do analyses for FGS even where the doctor has not thought about FGS as a differential diagnosis, i.e., not specifically requested by the clinician. Specific training is needed for these groups, e.g., when asked to analyse for Chlamydia or gonorrhoea, laboratory personnel could suggest to the clinician that Schistosoma polymerase chain reaction should be done [53].

**3.3.4 Ad hoc discussions on community awareness.** The participants emphasised the need for raising awareness in the community (amongst non-health professionals). They argued that the WHO recommends that each country should include FGS and community awareness in the Master Plan for Neglected Tropical Diseases. This is currently in the making in South Africa at the National Centre for Disease Control and Global Health Africa.

The three most common infectious diseases in Africa (HIV, tuberculosis (TB) and malaria) are known as "the Big 3". It was suggested that a slogan be created with "the Big 4", by adding FGS, giving the prevalence of the infectious disease and to promote its status as a public health problem in South Africa. This will also provide an opportunity to teach these 4 diseases together. In addition, it was pointed out that the SA National Department of Health had weekly prime-time slots allocated in the main television (TV) channel (every Wednesday 20:00–20:30 local time), which offered a window to local communities across the country. The use of easy-to-understand language, illustrations, and local analogies, noting the difference with a mosquito bite which can be felt as opposed to the FGS worm inadvertently penetrating the skin. These were strongly advised by all the groups. Finally, a TED Talk, comprising short

**Table 5. Summary of community level issues that need consideration.**

| Code | Category | Theme |
| --- | --- | --- |
| Schistosomiasis understood by general population in rural areas as a life milestone for men, but a stigma for women | Some people stay away from the rivers but then they may not learn how to swim | Social developmental consequences in low-resource settings |
| Children, parents and stakeholders (e.g, community nurse) unaware of FGS | FGS is not taught in schools nor there is awareness in the communities Didactic material, including videos and interactive exercises with peer-education accessible online/offline | Educational and social paths of FGS |
| Need for rising awareness interventions | Community awareness with targeted dissemination campaigns (e.g., the Big 4) based on previous successful strategies (e.g., Sesame Street for HIV and tuberculosis) | Proven rising awareness interventions |

video presentations should be spread via media channels (e.g., YouTube, WhatsApp, Facebook). Other suggestions included entertaining programs (such as the success of "Sesame Street" for raising awareness about HIV and TB), use of adverts in local newspapers, and artistically designed posters and leaflets that were easy to retain in memory.

A formal education curriculum for health professionals was suggested in parallel with work on the sociodemographic profile of the disease, addressed by campaigns targeting schools' programs and communities in FGS hot spot areas, in coordination with the departments of Health and Education. The desired outcomes of the teaching and learning of FGS focused on improving the current situation by focusing on education at community level. Formal education could include full lectures in schistosomiasis for first year students, ensuring that FGS is included. Reaching communities could be done in close collaborations with local communities' structures and the population in rural areas, spreading key messages via social media and other popular channels. Sanitation and recreational challenges such as "How to tell people they must not relieve themselves near or swim in the river nor use its water?" were agreed to be tackled with integrated approaches that involve policy makers (water treatment), educationalists and community leaders. In addition, the potential side effects of FGS treatment would have to be explained and understood by community leaders. They could have a specific role in treatment programs, such as encourage participation and monitor side effects. In the case of the WHO pocket atlas, community leaders were involved to be informed but side effects were monitored by the staff and local nurses.

A summary of the points mentioned above are shown in Table 5.

## 3.4 Workshop discussion points

**3.4.1 Myths and community perceptions.** One participant described that he had admired the "big boys" in his school and looked forward to the day that he would be wincing at the end of urination, just like them–however, that is a sign of urinary schistosomiasis (terminal dysuria). According to the participants, haematuria is easy to recognise in males during urination. However, girls squat to urinate, and they may not see the red colour of the urine. Furthermore, other symptoms of schistosomiasis in females may be misinterpreted e.g., if they have abnormal vaginal bleeding or painful intercourse, in young children there may be a suspicion of sexual abuse.

## 4. Discussion

This research has presented three co-design processes connected to FGS, an NTD that can be controlled or eliminated through effective interventions [47]. One of the main insurmountable problems related to FGS is that, in endemic areas, health professionals do not currently

diagnose nor are trained on FGS. The disease is simply overlooked, patients wrongly treated and misinformed. Millions live with the social and psychological repercussions of this misinformation, the physical pain, bleeding, and malodour. The complications of the disease include contact bleeding during gynaecological examination, infertility, abortion/ectopic pregnancy, involuntary urination produced by sudden movements (e.g., cough, laugh, jumps), genital ulcers, tumour and swellings in vulva, vagina, and cervix. Traditional research models that are "pathogen-focused" have had limited success focusing mainly on the mass drug administration (MDA), a vertical approach, because they did not consider the knowledge of local stakeholders, norms, and sociocultural, historical, political, economic, and ecological contexts of the people to whom the research, program or intervention was targeted [37]. Studies have found that FGS knowledge was lacking in front-line health workers, girls, and women, with sexual promiscuity and STI treatment among the general misconceptions of FGS [41]. Therefore, the knowledge of a community's perception about interventions to control schistosomiasis can be valuable to policy makers and program implementers intending to set up interventions co-managed by the community members [54].

This research has applied a so-called peer model, HCD, facilitating the emerging of value from local stakeholders' knowledge, who provide a rich insight and experience regarding the context where FGS is endemic. These models seem to be most successful when they are focused on ongoing commitment to collaboration, co-learning, mutual leadership, and equitable partnership through shared decision-making [55]. A participatory process aimed at ideating solutions to address stakeholders' needs, created for and by them [37, 56, 57].

The process supported co-design workshops whose outcomes had different degree of finalisation, but all made with the direct involvement of key stakeholders: gynaecologists, nurses, Higher Education teachers, health students, citizens, national and world authorities, and FGS world experts. The main outcome of the first workshop was the production of the WHO FGS Atlas prototype depicting lesions characteristic of FGS, followed by a poster with lesion examples, presentations for promotion and training and a dedicated webpage. The goal of the FGS Atlas was to enable nurses and doctors to be informed of FGS, to diagnose it or refer to others who can, and to inform their patients. Increasing the index of suspicion will allow for a preoperative diagnosis of FGS, avoidance of unnecessary radical surgery, and misdiagnosis as STIs. Furthermore, an effective diagnosis of FGS will help women make sexual health choices that may decrease their risk of HIV [10, 58]. The poster and the focus on the promotion, dissemination and transfer of knowledge respond to the need for improving drug compliance based on intensive health education campaigns. These provide disease specific information, but also deal with prevailing misconceptions about transmission, prevention, treatment, and drug side-effects before conducting MDA [59].

The main outcomes of the second and third workshops were the generation and selection of knowledge for an e-learning course to instruct those who will diagnose patients using speculums and/or colposcope; a discussion of the potential curriculum for future health professionals; and a low-fidelity prototype for a diagnostic support system for FGS in low-resource settings. Firstly, participants' interventions helped to provide information for researchers about the ecology of the disease. Participants corroborated the common understanding in rural areas that FGS is a life stage in the case of men (maturity milestone), and stigmatisation in the case of women (interpreted as a symptom of a STI). In addition, they exemplified the fundamental role that the ecosystem of transmission, the water, plays in many communities. These reports are in line with gaps found elsewhere [60] that identified sociocultural barriers towards the uptake of preventive and treatment services in Sub-Saharan Africa. Gaps in transmission control and treatment have been previously reported in endemic areas, urging provision of interventions for the education of students and teachers in endemic areas to be able to

recognise water contact activity risk, improve participation in MDA and promote interest in science in a practical way [61]. Therefore, the focus on the education and training in the workshops aligns with the pathways pointed out by the literature, where a comprehensive health education program using contextual and standardised training tools may improve peoples' knowledge, attitudes, and practices in relation to schistosomiasis prevention and control. The findings also highlight the significance of including caregivers in the planning and implementation of schistosomiasis control programs [60]. Investing in the education of community health workers with appropriate disease-specific training is crucial if disease elimination is ultimately to be achieved [59]. Educators from all levels (from pre-school children's schools to Higher Education institutions) can be influential and effective agents of change in local and wider communities. They can help to expand and increase acceptance of elimination activities, and therefore contribute to community-level behavioural intervention. They might constitute an unused resource that can help to expand and increase acceptance of and participation in schistosomiasis and other NTD control activities in Africa [62]. These findings highlight the capacity of local stakeholders to be part of interventions that contribute to schistosomiasis control programs [63]. The diagnostic support system of FGS in low-resource settings addresses the need for new diagnostic tests to help in defining elimination. An integrated approach with political support and community engagement are essential to have a meaningful and long-lasting effect in the communities of patients, health professionals and citizens. This approach is in line with calls for multi-sectoral approaches to move from control to elimination of schistosomiasis in sub-Saharan Africa [64].

One of the strengths of the outcomes of the research presented here is that it is already influencing the clinical practice in the field of FGS in the Sub-Saharan Africa via the WHO FGS Atlas. This has been distributed to 51 anglophone and francophone African countries. In addition, this research presented evidence of the successful implementation of a peer-model method that led to the co-creation of such an Atlas. However, the workshops continued to provide insights by going further and addressing urgent calls for education, training, digitalisation, and diagnosis support. Even at this early stage, the outcomes shed light on key components of such developments, such as who (front-line health workers and professionals, health students, educators, authorities), when (now), how (peer-model) need to be involved in the endeavour of prevention, treatment, and eradication of FGS. Low-cost training approaches such as the training of health professionals who use a speculum are considered instrumental in the approach of appropriate diagnosis of FGS in the field.

HCD stems from the assessment of what is happening on the ground as voiced by members of the community and other stakeholders involved in health and care in such communities. The HCD peer-model enables through participation and representativeness, transformative research, and evaluation, aiming at reducing the distance between those who design and implement a research project and those who are its recipients. In other words, intervening in existing practices (or absence of them) and established social systems requires those who are the target of such interventions to take part in how such intervention is formulated, conducted, and concluded [65], aligned with indigenous research paradigms ("nothing about us without us" [66]).

The conclusions add to the existing knowledge in the field by presenting evidence of a viable method to produce a valuable aid for health workers, together with a roadmap for the next steps to take into consideration for education, training, and diagnosis support. Future research can build upon the outcomes here presented, following up in the reported challenges (i.e., how to obtain high-quality images from the whole vaginal walls; effective dissemination campaigns). Key milestones on the way ahead include the involvement and ownership of such initiatives by the same communities worst affected by FGS, together with local, regional, national,

and international authorities in harmony with FGS initiatives [41]. These communities frequently have limited access without alternatives to water sources, regularly facing schistosomiasis risk. Therefore, national control programs need "to appropriately couple preventive services with sexual and reproductive primary health care with future training of health workers for improved management of FGS cases" [67]. Integrated strategies should implement public health measures, such as improved access to clean water and sanitation, promotion of behavioural changes and wide access treatment with praziquantel covering all age groups to reach interruption of schistosome transmission and elimination goals [68]. However, when public health strategies solely focus on technical solutions intended to be homogeneously implemented across different contexts, the social aspects that significantly shape the developments of such programs are also neglected. Only an evidence-based approach that takes into consideration both the biological and social dynamics can support a sustainable control of NTDs [69]. This approach requires an understanding of the human body, and therefore of disease, which both interact with local environments and become indistinct from other aspects of culture (the concept of "local biologies"; from [70, 71]. Thus, it is imperative to overcome the many challenges ahead (e.g., theoretical, methodological, cultural, political) for the fruitful control of NTDs [72].

### 4.1 Clinical relevance

The limitations lie in the clinical efficacy and reproducibility of this research. Even though the WHO FGS Atlas has been a significant improvement in the collection and dissemination of visual and clinical information in a didactic way, there still are challenges to address. For instance, the Atlas must be understood as an instrument for appropriate diagnosis of FGS. This depends on the willingness and information that each health worker and professional, together with their leadership, have about FGS and its impact in the society. Therefore, the existence and distribution of the Atlas does not directly eliminate neglect in clinical practice without support, and more research is needed in these aspects. The reproducibility of the HCD method in the times of a pandemic (e.g., Covid-19), reduces the possibility of recruiting and gathering different stakeholders to co-design around a table. Therefore, low-cost digital means would be necessary to repeat such process until a medicine and/or a vaccine is widely available for the general population.

### 4.2 Limitations on the data analysis approach

There are several limitations related to this research. Firstly, the quality of several images was not sufficient or optimal to show observable lesions (as mentioned in the text), therefore they were discarded. A more technically advanced approach regarding image quality could have been performed, although the one chosen was considered practical for the resources available. The sample size, interpreted as the number of workshop participants may have been considerable higher. However, two of the criteria (already mentioned) was the representativity and diversity of the stakeholders. Thus, these latter factors may have contributed to a better balance between number and stakeholder representation. There were few if any assumptions present in the research, although researchers might always be blind on their own practices. All the statements provided in the manuscript have been rigorously backed up by evidence or at least discussed alternatives when otherwise. Finally, the methods employed, HCD, qualitative data analysis, always face the challenge of introducing objectivity from a subjective perspective. Here, the ethos has been the equitable research partnership with local stakeholders, trying to reach a fair representation but most and foremost an equitable distribution of knowledge.

### 4.3 Lessons learned from the implementation and the roll-out of the WHO pocket atlas 9 years after. A way forward

The research here presented is the result of a two-decade long research on FGS that allowed for the publication of the WHO pocket atlas in 2014. The value of the method employed for the design and implementation of the WHO pocket atlas serves as a model for new diseases without an established management protocol. Among the lessons learned from its dissemination, the WHO pocket atlas was not found in the local schistosomiasis endemic clinics visited by the research team in 4 of the 53 African countries. Lastly, the poster was not printed in large scale due to its big size, which led to the imprint of just the specifications.

This publication was substantially delayed by the COVID-19 pandemic, that struck in March 2020. However, that time allowed the authors to review the process and the material collected, becoming an essential part of a research proposal submitted to the European Union in 2021. The proposal was awarded 8 million euros for the project called "DUALSAVE-FGS, Dual diagnosis by Spectral Artificial Visual Examination for Female Genital Schistosomiasis and cervical cancer. Digital, new, low-cost, and simple diagnosis and training" (Grant agreement ID: 101057853).

## Supporting information

**S1 Checklist. A supporting information file with the inclusivity in global research form is attached.**
(DOCX)

## Acknowledgments

We thank the women from endemic areas who participated in several African studies for the sake of their own health and for their sisters. We thank the scientists who made these studies possible (in alphabetical order): Baay MFD, Chitsulo L, Feldmeier H, Friis H, Gagai S, Gomo E, Gundersen SG, Gwanzura L, Hegertun IEA, Helling-Giese G, Holmen SD, Kjetland EF, Kleppa E, Kumwenda N, Kurewa EN, Kvalsvig JD, Lillebo K, Maphumulo A, Mason PR, Mduluza T, Midzi N, Ndhlovu PD, Ndung'u T, Norseth HM, Nyanga L, Onsrud M, Pillay P, Poggensee G, Rakotomanana F, Ramarokoto CE, Ramsuran V, Randjelovica A, Randrianasolo BS, Ravaoalimalala VE, Ravoniarimbinina P, Roald B, Sandvik L, Sjaastad A, Sulheim Gundersen KM, Taylor M, Ten Hove RJ, Thomassen Morgas DE, Van Lieshout L, Vennervald BJ, Zulu SG. The technical team, observers, support staff and PR team are thanked for working day and night before and during the workshop: Dr Pavitra Pillay, Dr Mari Molvik, Information officer Ida Amelie Helgesen, Siphosenkosi G Zulu, Kumud Nina Henriksen, Thulisile Mkhize, and John-Erik Armelius. The authors also thank Prof. Mariann Fossum, from the University of Agder, Norway, who substantially contributed to improve the quality of an early version of this manuscript.

## Author Contributions

**Conceptualization:** Santiago Gil Martinez, Pamela S. Mbabazi, Hashini N. Galaphaththi-Arachchige, Eyrun F. Kjetland.

**Data curation:** Santiago Gil Martinez, Eyrun F. Kjetland.

**Formal analysis:** Sibone Mocumbi, Sigve D. Holmen, Eyrun F. Kjetland.

**Funding acquisition:** Eyrun F. Kjetland.

**Investigation:** Santiago Gil Martinez, Sibone Mocumbi, Hashini N. Galaphaththi-Arachchige, Sigve D. Holmen, Eyrun F. Kjetland.

**Methodology:** Santiago Gil Martinez, Pamela S. Mbabazi, Sibone Mocumbi, Hashini N. Galaphaththi-Arachchige, Tsakani Furumele, Eyrun F. Kjetland.

**Project administration:** Eyrun F. Kjetland.

**Resources:** Eyrun F. Kjetland.

**Software:** Eyrun F. Kjetland.

**Supervision:** Eyrun F. Kjetland.

**Validation:** Eyrun F. Kjetland.

**Visualization:** Eyrun F. Kjetland.

**Writing – original draft:** Santiago Gil Martinez.

**Writing – review & editing:** Santiago Gil Martinez, Pamela S. Mbabazi, Motshedisi H. Sebitloane, Bellington Vwalika, Sibone Mocumbi, Hashini N. Galaphaththi-Arachchige, Sigve D. Holmen, Bodo Randrianasolo, Borghild Roald, Femi Olowookorun, Francis Hyera, Sheila Mabote, Takalani G. Nemungadi, Thembinkosi V. Ngcobo, Tsakani Furumele, Patricia D. Ndhlovu, Martin W. Gerdes, Svein G. Gundersen, Zilungile L. Mkhize-Kwitshana, Myra Taylor, Roland E. E. Mhlanga, Eyrun F. Kjetland.

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
