## [Decision Letter · Decision Letter 0]

3 May 2023

PGPH-D-23-00298

The WHO Atlas for Female-genital Schistosomiasis: co-design of a practicable diagnostic guide, digital support and training

Dear Dr. Martinez,

Thank you for submitting your manuscript to PLOS Global Public Health. After careful consideration, we feel that it has merit but does not fully meet PLOS Global Public Health’s publication criteria as it currently stands. Therefore, we invite you to submit a revised version of the manuscript that addresses the points raised during the review process.

We look forward to receiving your revised manuscript.

Kind regards,

Ateeb Ahmad Parray, BDS, MSS, MPH

Academic Editor

Journal Requirements:

1. In the ethics statement in the Methods, you have specified that verbal consent was obtained. Please provide additional details regarding how this consent was documented and witnessed, and state whether this was approved by the IRB.

2. Please amend your detailed online Financial Disclosure statement. This is published with the article. It must therefore be completed in full sentences and contain the exact wording you wish to be published.

a) State the initials, alongside each funding source, of each author to receive each grant. For example: "This work was supported by the National Institutes of Health (####### to AM; ###### to CJ) and the National Science Foundation (###### to AM)."

3. Please ensure that the funders and grant numbers match between the Financial Disclosure field and the Funding Information tab in your submission form. Note that the funders must be provided in the same order in both places as well.

4. Please update your online Competing Interests statement. If you have no competing interests to declare, please state: “The authors have declared that no competing interests exist.”

5. Please ensure that you refer to Figure 7 in your text as, if accepted, production will need this reference to link the reader to the figure.

6. Please include a legend for Figure 7 in your manuscript.

7. Please ensure that you refer to Tables 3, 4 and 5 in your text as, if accepted, production will need these references to link the reader to the tables.

8. Some material included in your submission may be copyrighted. According to PLOS’s copyright policy, authors who use figures or other material (e.g., graphics, clipart, maps) from another author or copyright holder must demonstrate or obtain permission to publish this material under the Creative Commons Attribution 4.0 International (CC BY 4.0) License used by PLOS journals. Please closely review the details of PLOS’s copyright requirements here: PLOS Licenses and Copyright. If you need to request permissions from a copyright holder, you may use PLOS's Copyright Content Permission form.

Potential Copyright Issues:

Figures 1, 2, 3, 4 and 8 includes an image of an identifiable person. Please provide written confirmation or release forms, signed by the subject(s) (or their parent/legally authorized guardian), giving permission to be photographed and to have their images published under our CC-BY 4.0 license. 

Otherwise, we kindly request that you remove the photograph.

Figures 4 and 8 contains branding/a logo. We are not permitted to publish this under our CC-BY 4.0 license, even with permission. We ask that you please remove or replace it.

Additional Editor Comments (if provided):

Reviewers' comments:

Reviewer's Responses to Questions

**Comments to the Author**

1. Does this manuscript meet PLOS Global Public Health’s publication criteria? Is the manuscript technically sound, and do the data support the conclusions? The manuscript must describe methodologically and ethically rigorous research with conclusions that are appropriately drawn based on the data presented.

Reviewer #1: Yes

Reviewer #2: Yes

2. Has the statistical analysis been performed appropriately and rigorously?

Reviewer #1: N/A

Reviewer #2: N/A

3. Have the authors made all data underlying the findings in their manuscript fully available (please refer to the Data Availability Statement at the start of the manuscript PDF file)?

Reviewer #1: Yes

Reviewer #2: Yes

4. Is the manuscript presented in an intelligible fashion and written in standard English?

Reviewer #1: Yes

Reviewer #2: Yes

5. Review Comments to the Author

Reviewer #1: PLOS Global Public Health: PGPH-D-23-00298

The WHO Atlas for Female-genital Schistosomiasis: co-design of a practicable diagnostic guide, digital support and training

(Short Title: The WHO Atlas for Female-genital Schistosomiasis)

(Recommend accept with minor revisions)

This is an extremely important topic, where although the presence of Female-genital Schistosomiasis (FGS) has been recognised for many years (as pointed out in the manuscript), it has only relatively recently attained global attention. The WHO atlas for FGS described herein is a fantastic resource and represents a major contribution to the field of genital schistosomiasis (in terms of awareness of the condition and clinical diagnosis).

This study presents original findings from three multi-disciplinary workshops performed 2014, and twice in 2019. The workshops were centred around the method of ‘co-design’ defined as the ‘creativity of designers and people not trained in design, working together in the design development process’, being particularly useful within the context of healthcare providers ‘experience-based co-design’. The workshops are carefully described, and outcomes defined. It is very difficult to now quantify and assimilate these results, however the atlas has been widely distributed and commended (anecdotally) since its publication.

General comments:

It would be interesting to know within the manuscript how many participants had colposcopy performed and how many images were originally obtained in order to select representative samples (the atlas itself suggests 10,000 but this is not discussed within the manuscript).

The first of the workshops was held on 04/10/2014, designing the atlas (WHO published in 2015). As it is now 2023, I feel the authors should discuss (or at least acknowledge) the extensive time-lag in publishing these data. Since it has been several years – do the authors have any data on the implementation of the atlas +/- the corresponding poster (note the lack of data would not preclude publication)?

It would be useful to include the web address of the WHO atlas download page:

Female genital schistosomiasis: A pocket atlas for clinical health-care professionals (who.int)

The majority of language used is clear and written in standard English. The workshops and atlas design were performed to a high ethical standard; submitted to several ethical regulatory bodies in different organizations and countries.

 

Specific comments:

Comment 1:

Page 13 first sentence: “Before the workshop, two FGS experts (EFK and HGA) sorted images by lesion type and quality and selected 28 images were chosen for each of the four FGS lesion types.”

• This sentence is ambiguous as it seems to be reporting 28 images were pre-selected (but I’m aware only 28 images were included in the final atlas)

Page 15 second sentence: “The experts had pre-selected 60 images of the types of lesions of FGS.”

• Please clarify how many images were pre-selected.

Comment 2:

Table 3 page 20: It is not clear what 'Categories 1 and 2' represent? Please include a description in the table and/or legend.

Table 5 page 27: Please state what the 2x question marks (?) along with the 'Category' represent.

It may be useful to include a content analysis summary table with HCD-phases for the first workshop, atlas design. It would also be clearer to have content analysis summary tables displaying consistent column headings e.g., HCD Phase (column heading on Table 3 but not 4).

Reviewer #2: This WHO Pocket Atlas for FGS article is a valuable contribution to the field of neglected tropical diseases that afflict merginalized populations. The paper is well-written, and the presented data are both persuasive and instructive. This paper is a vital tool for researchers, healthcare professionals, and policymakers working on the development of the WHO Pocket Atlas, which can be readily utilized by health professionals in low-resource settings. Additionally contribute to the prevention and management of Female-genital Schistosomiasis.

6. PLOS authors have the option to publish the peer review history of their article (what does this mean?). If published, this will include your full peer review and any attached files.

**Do you want your identity to be public for this peer review?** For information about this choice, including consent withdrawal, please see our Privacy Policy.

Reviewer #1: No

Reviewer #2: No

---

## [Decision Letter · Decision Letter 1]

18 Sep 2023

PGPH-D-23-00298R1

The WHO Atlas for Female-genital Schistosomiasis: co-design of a practicable diagnostic guide, digital support and training

Dear Dr. Martinez,

Thank you for submitting your manuscript to PLOS Global Public Health. After careful consideration, we feel that it has merit but does not fully meet PLOS Global Public Health’s publication criteria as it currently stands. Therefore, we invite you to submit a revised version of the manuscript that addresses the points raised during the review process.

We look forward to receiving your revised manuscript.

Kind regards,

Ateeb Ahmad Parray, BDS, MSS, MPH

Academic Editor

Journal Requirements:

1. In the ethics statement in the Methods, you have specified that verbal consent was obtained. Please provide additional details regarding how this consent was documented and witnessed, and state whether this was approved by the IRB"

2. Please include a complete copy of PLOS’ questionnaire on inclusivity in global research in your revised manuscript. Our policy for research in this area aims to improve transparency in the reporting of research performed outside of researchers’ own country or community. The policy applies to researchers who have travelled to a different country to conduct research, research with Indigenous populations or their lands, and research on cultural artefacts. The questionnaire can also be requested at the journal’s discretion for any other submissions, even if these conditions are not met.  Please find more information on the policy and a link to download a blank copy of the questionnaire here: https://journals.plos.org/globalpublichealth/s/best-practices-in-research-reporting. Please upload a completed version of your questionnaire as Supporting Information when you resubmit your manuscript.

Additional Editor Comments (if provided):

Reviewers' comments:

Reviewer's Responses to Questions

**Comments to the Author**

1. If the authors have adequately addressed your comments raised in a previous round of review and you feel that this manuscript is now acceptable for publication, you may indicate that here to bypass the “Comments to the Author” section, enter your conflict of interest statement in the “Confidential to Editor” section, and submit your "Accept" recommendation.

Reviewer #3: (No Response)

Reviewer #4: (No Response)

2. Does this manuscript meet PLOS Global Public Health’s publication criteria? Is the manuscript technically sound, and do the data support the conclusions? The manuscript must describe methodologically and ethically rigorous research with conclusions that are appropriately drawn based on the data presented.

Reviewer #3: Partly

Reviewer #4: Partly

3. Has the statistical analysis been performed appropriately and rigorously?

Reviewer #3: Yes

Reviewer #4: N/A

4. Have the authors made all data underlying the findings in their manuscript fully available (please refer to the Data Availability Statement at the start of the manuscript PDF file)?

Reviewer #3: Yes

Reviewer #4: Yes

5. Is the manuscript presented in an intelligible fashion and written in standard English?

Reviewer #3: Yes

Reviewer #4: Yes

6. Review Comments to the Author

Reviewer #3: Overall the manuscript will be much improved once the authors' have addressed the majority of the reviewers' concerns comprehensively. Just a bit more clarity/detail for the few points and in certain sections, for e.g. ethics or IRB review would further strengthen it. Nice work!

Reviewer #4: The manuscript has additional revisions that should be addressed prior to publication.

7. PLOS authors have the option to publish the peer review history of their article (what does this mean?). If published, this will include your full peer review and any attached files.

**Do you want your identity to be public for this peer review?** For information about this choice, including consent withdrawal, please see our Privacy Policy.

Reviewer #3: **Yes: **Khushbu Balsara

Reviewer #4: No

---

## [Editor Report · Decision Letter 2]

31 Oct 2023

PGPH-D-23-00298R2

The WHO Atlas for Female-genital Schistosomiasis: co-design of a practicable diagnostic guide, digital support and training

Dear Dr. Martinez,

Thank you for submitting your manuscript to PLOS Global Public Health. After careful consideration, we feel that it has merit but does not fully meet PLOS Global Public Health’s publication criteria as it currently stands. Therefore, we invite you to submit a revised version of the manuscript that addresses the points raised during the review process.

We look forward to receiving your revised manuscript.

Kind regards,

Ateeb Ahmad Parray, BDS, MSS, MPH

Academic Editor
---

## [Decision Letter · Decision Letter 3]

12 Dec 2023

PGPH-D-23-00298R3

The WHO Atlas for Female-genital Schistosomiasis: co-design of a practicable diagnostic guide, digital support and training

Dear Dr. Martinez,

Thank you for submitting your manuscript to PLOS Global Public Health. After careful consideration, we feel that it has merit but does not fully meet PLOS Global Public Health’s publication criteria as it currently stands. Therefore, we invite you to submit a revised version of the manuscript that addresses the points raised during the review process.

We look forward to receiving your revised manuscript.

Kind regards,

Ateeb Ahmad Parray, BDS, MSS, MPH

Academic Editor

Journal Requirements:

1. In the ethics statement in the Methods, you have specified that verbal consent was obtained. Please provide additional details regarding how this consent was documented and witnessed, and state whether this was approved by the IRB"

2. Please include a complete copy of PLOS’ questionnaire on inclusivity in global research in your revised manuscript. Our policy for research in this area aims to improve transparency in the reporting of research performed outside of researchers’ own country or community. The policy applies to researchers who have travelled to a different country to conduct research, research with Indigenous populations or their lands, and research on cultural artefacts. The questionnaire can also be requested at the journal’s discretion for any other submissions, even if these conditions are not met.  Please find more information on the policy and a link to download a blank copy of the questionnaire here: https://journals.plos.org/globalpublichealth/s/best-practices-in-research-reporting. Please upload a completed version of your questionnaire as Supporting Information when you resubmit your manuscript.

Additional Editor Comments (if provided):

Reviewers' comments:

Reviewer's Responses to Questions

**Comments to the Author**

1. If the authors have adequately addressed your comments raised in a previous round of review and you feel that this manuscript is now acceptable for publication, you may indicate that here to bypass the “Comments to the Author” section, enter your conflict of interest statement in the “Confidential to Editor” section, and submit your "Accept" recommendation.

Reviewer #3: (No Response)

2. Does this manuscript meet PLOS Global Public Health’s publication criteria? Is the manuscript technically sound, and do the data support the conclusions? The manuscript must describe methodologically and ethically rigorous research with conclusions that are appropriately drawn based on the data presented.

Reviewer #3: No

3. Has the statistical analysis been performed appropriately and rigorously?

Reviewer #3: N/A

4. Have the authors made all data underlying the findings in their manuscript fully available (please refer to the Data Availability Statement at the start of the manuscript PDF file)?

Reviewer #3: Yes

5. Is the manuscript presented in an intelligible fashion and written in standard English?

Reviewer #3: Yes

6. Review Comments to the Author

Reviewer #3: I kindly request your attention to the earlier comments that haven't been addressed yet. Additionally, it would be more respectful to avoid stigmatizing terms like "not virgin women." Using "sexually active women" is more appropriate and sufficient.

7. PLOS authors have the option to publish the peer review history of their article (what does this mean?). If published, this will include your full peer review and any attached files.

**Do you want your identity to be public for this peer review?** For information about this choice, including consent withdrawal, please see our Privacy Policy.

Reviewer #3: **Yes: **Khushbu Balsara

---

## [Editor Report · Decision Letter 4]

31 Jan 2024

The WHO Atlas for Female-genital Schistosomiasis: co-design of a practicable diagnostic guide, digital support and training

PGPH-D-23-00298R4

Dear Dr Martinez,

We are pleased to inform you that your manuscript 'The WHO Atlas for Female-genital Schistosomiasis: co-design of a practicable diagnostic guide, digital support and training' has been provisionally accepted for publication in PLOS Global Public Health.

Best regards,

Ateeb Ahmad Parray, BDS, MSS, MPH

Academic Editor